

# *Nurhachius luei,* a new istiodactylid pterosaur (Pterosauria, Pterodactyloidea) from the Early Cretaceous Jiufotang Formation of Chaoyang City, Liaoning Province (China) and comments on the Istiodactylidae

Xuanyu Zhou[1,2], Rodrigo V. Pêgas[3], Maria E.C. Leal[4,5] and Niels Bonde[5,6]

[1] Institute of Geology, Chinese Academy of Geological Sciences, Beijing, China
[2] China University of Geosciences, Beijing, China
[3] Laboratory of Vertebrate Paleontology and Animal Behavior, Universidade Federal do ABC, São Bernardo, São Paulo, Brazil
[4] Departamento de Geologia, Universidade Federal do Ceará, Fortaleza, Ceará, Brazil
[5] Zoological Museum (SNM), Copenhagen University, Copenhagen, Denmark
[6] Fur Museum (Museum Saling), Fur, Denmark

## ABSTRACT

A new istiodactylid pterosaur, *Nurhachius luei* sp. nov., is here reported based on a complete skull with mandible and some cervical vertebrae from the lower part of the Jiufotang Formation of western Liaoning (China). This is the second species of *Nurhachius*, the type-species being *N. ignaciobritoi* from the upper part of the Jiufotang Formation. A revised diagnosis of the genus *Nurhachius* is provided, being this taxon characterized by the presence of a slight dorsal deflection of the palatal anterior tip, which is homoplastic with the Anhangueria and *Cimoliopterus*. *N. luei* sp. nov. shows an unusual pattern of tooth replacement, with respect to other pterodactyloid species. The relationships within the Istiodactylidae and with their closest taxa are investigated through a phylogenetic analysis by parsimony.

## INTRODUCTION

Istiodactylid pterosaurs are characterized by rhombic teeth with lancet-shaped crowns, long skulls with short pre-antorbital portions of the rostrum, and nasoantorbital fenestrae representing over 50% of the total skull length and height (*Howse, Milner & Martill, 2001*; *Andres & Ji, 2006*; *Lü et al., 2013*). The group was originally named by *Howse, Milner & Martill (2001)* in order to accommodate, then, only *Istiodactylus latidens*. Later, the Istiodactylidae were phylogenetically defined by *Andres, Clark & Xu (2014)* as the least inclusive clade containing *Nurhachius* and *Istiodactylus*.

Four pterosaur genera and five species (all represented by a single specimen) have been referred to the Istiodactylidae sensu *Andres, Clark & Xu (2014)* in the literature, namely

Corresponding authors
Xuanyu Zhou, zhouxy2017@yeah.net
Rodrigo V. Pêgas,
rodrigo.pegas@hotmail.com

*Istiodactylus latidens, Istiodactylus* sinensis*, Liaoxipterus brachyognathus, Nurhachius ignaciobritoi,* and *Longchengpterus zhaoi.* However, *Longchengpterus zhaoi* has been considered a junior synonym of *N. ignaciobritoi* by *Lü, Xu & Ji (2008)*, a view that is followed here (see Discussion). Therefore, *N. ignaciobritoi* is the only Chinese istiodactylid species to be represented by two specimens so far. *Haopterus gracilis, Hongshanopterus lacustris,* and *Archaeoistiodactylus linglongtaensis* have been reported in literature as taxa that are close to the Istiodactylidae (*Wang & Lü, 2001*; *Wang et al., 2008a*; *Lü & Fucha, 2010*). However, the affinity of *Archaeoistiodactylus linglongtaensis* has been questioned by *Sullivan et al. (2014)*.

All istiodactylid pterosaurs are from the Early Cretaceous Jiufotang Formation of northeastern China with the exception of *Istiodactylus latidens*, which is from the Early Cretaceous Vectis Formation of the Isle of Wight, Southern England. Also, the three taxa that are reported as close to istiodactylids come from northeastern China and surrounding areas: *Haopterus gracilis* is from the Early Cretaceous Yixian Formation, *Hongshanopterus lacustris* from the Jiufotang Formation, and *Archaeoistiodactylus linglongtaensis* from the Middle Jurassic Tiaojishan Formation. Apart from the latter, these Chinese pterosaurs belong to the Jehol Biota (see *Chang et al., 2003*).

By the end of 2016, 23 species of pterosaurs from the Jiufotang Formation have been reported (*Andres & Ji, 2006*; *Zhiming & Junchang, 2005*; *Dong, Sun & Wu, 2003*; *Jiang et al., 2016*; *Rodrigues et al., 2015*; *Li, Lü & Zhang, 2003*; *Lü & Ji, 2005*; *Lü & Yuan, 2005*; *Lü et al., 2006*, *2007*, *2016a*, *2016b*; *Lü, Xu & Ji, 2008*; *Wang & Zhou, 2003a*, *2003b*; *Wang et al., 2005*, *2006*, *2008a*, *2008b*, *2012*, *2014*).

In this paper, we describe a second species of *Nurhachius* from the Jiufotang Formation and investigate the phylogenetic relationships of the istiodactylids and purported close taxa.

## Geological, paleontological and geochronological information

The Jiufotang Formation is known worldwide for its paleontological richness and the exquisite preservation of its fossils, which include plants, insects, fishes, mammals, birds, non-avian dinosaurs, and pterosaurs (*Wang, 2018*; *Meng, Wang & Li, 2011*; *Wang & Zhou, 2019*; *Yao et al., 2019*). Fossils occur mainly in the lower part of the formation, known as Boluochi Beds or Boluochi Member (see *Chang et al., 2003*), which is characterized by the *Jinanichthys—Cathayornis* Fauna that includes small feathered dinosaurs like the four-winged *Microraptor* (*Xu et al., 2003*) and several pterosaurs (*Chang et al., 2003*; *Zhou, Barrett & Hilton, 2003*).

The Jiufotang Formation is 206–2,685 m thick according to *Chang et al. (2009)* and is mainly composed of mudstone, siltstone, shale, sandstone and tuff. A tuff from the basal part of formation (two m above the boundary between the Yixian and Jiufotang formations) in western Liaoning was dated to 122.1 ± 0.3 Ma by *Chang et al. (2009)*. A basalt in the upper part of the formation in Inner Mongolia was dated to 110.59 ± 0.52 Ma by *Eberth et al. (1993)*, but *Chang et al. (2009)* objected that the correlation between the Jiufotang Formation in Liaoning and Inner Mongolia is unclear and the age of the uppermost Jiufotang Formation remains unknown. The Aptian age of the Early

Cretaceous ranges ~125–113 Ma according to the International Chronostratigraphic Chart 2018/08. Therefore, the Jiufotang Formation is Aptian in age, but might reach the Albian.

The Jiufotang Formation and the underlying Yixian Formation traditionally constitute the Jehol Group. The Yixian Formation is 225–4,000 m thick, varying in thickness and lithology in different areas according to *Chang et al. (2009)*, but only a fraction is made of sedimentary rocks because basalts and lavas represents a substantial part of the section. *Chang et al. (2009)* dated the basal part of the Yixian Formation in Western Liaoning to 129.7 ± 0.5 and the uppermost part of the underlying Tuchengzi Formation to 139.5 ± 1.0 Ma. The upper part of the Yixian Formation (the Jingangshan Beds) was dated to 126.5 Ma (*Chang et al., 2003*). Therefore, the Yixian Formation represents an interval of ~7 Ma from early Barremian to early Aptian and the Jiufotang Formation might represents an interval of over 11 Ma from early Aptian to early Albian.

The Jehol Group has yielded the famous Jehol biota. Four fossil-bearing levels with partly different fossil associations have been distinct within the Yixian Formation and only one (corresponding to the Boluochi Beds) in the Jiufotang Formation (*Chang et al., 2003*).

Both the holotype of *N. ignaciobritoi* and its referred specimen (the holotype of *Longchengpterus zhaoi*) come from the upper part of the Jiufotang Formation (see *Wu et al., 2018*), whereas the new species comes from the Boluochi Beds (lower part of the Jiufotang Formation).

## MATERIALS AND METHODS

The holotype and only specimen of the new species consists of a skull with mandible and seven articulated cervical vertebrae. It was previously figured in *Lü et al. (2013*, figures at pp. 81–82*)* and reported as an unnamed istiodactylid. The specimen was found near the village of Huanghuatan (Dapingfang town, Chaoyang City, western Liaoning).

For the comparisons we present, the following taxa/specimens were analyzed first-hand (by XZ): *N. ignaciobritoi* (both specimens, LPM 00023 and IVPP V-13288), *Liaoxipterus brachyognathus* holotype (CAR-0018), *Hongshanopterus lacustris* holotype (IVPP V14582) and *Haopterus gracilis* holotype (IVPP V11726). Data from other taxa was gathered from the literature.

A phylogenetic analysis was performed based on the data matrix by *Holgado et al. (2019)* modified with the inclusion of characters by *Lü, Xu & Ji (2008)*, *Witton (2012)*, *Andres, Clark & Xu (2014)*, and new characters; and the addition of the following taxa: *Archaeoistiodactylus linglongtaensis*, *Kunpengopterus sinensis*, *Liaoxipterus brachyognathus* and *N. luei* sp. nov. (see Supplemental Information). The analysis was performed by TNT (*Goloboff, Farris & Nixon, 2008*) using the Traditional Search option, 10,000 replicates, random seed = 0 and collapsing trees after search. The character and character states list and the TNT file with the data matrix are available in the Supplemental Information.

The electronic version of this article in portable document format will represent a published work according to the International Commission on Zoological Nomenclature (ICZN), and hence the new names contained in the electronic version are effectively published under that Code from the electronic edition alone. This published work and the

nomenclatural acts it contains have been registered in ZooBank, the online registration system for the ICZN. The ZooBank LSIDs (Life Science Identifiers) can be resolved and the associated information viewed through any standard web browser by appending the LSID to the prefix http://zoobank.org/. The LSID for this publication is: urn:lsid:zoobank.org:pub:03EF173E-4AB5-4C74-B80C-A6AAFA65E61C. The online version of this work is archived and available from the following digital repositories: PeerJ, PubMed Central, and CLOCKSS.

## RESULTS

### Systematic Paleontology

Pterosauria *Kaup, 1834*
Pterodactyloidea *Plieninger, 1901*
Istiodactylidae *Howse, Milner & Martill, 2001* (sensu *Andres, Clark & Xu, 2014*)
*Nurhachius* Wang et al., 2005

**Type species.** *Nurhachius ignaciobritoi* Wang et al., 2005

**Synonym.** *Longchengpterus zhaoi* Wang et al., 2006

**Emended Diagnosis.** Istiodactylids that share the following features: slight dorsal deflection of the palatal anterior tip; orbit piriform; craniomandibular joint located under the anterior margin of the orbit; dentary symphysis about one-third the length of the mandible; dentary symphysis with gradual taper of the lateral margins; triangular, laterally compressed teeth lacking carinae; crowns with both labial and lingual slight concavities; slight constriction between tooth crown and root.

*Nurhachius luei* sp. nov.

**ZooBank LSID for species.** urn:lsid:zoobank.org:act:6F93DC7F-20A7-4CBC-8A38-1D6C802A1906.

**Etymology.** The specific name *luei* (/lyi/) honors the late Prof. Junchang Lü, who has made great contributions to the study of Chinese pterosaurs.

**Holotype.** Skull, mandible and seven cervical vertebrae (BPMC-0204). The specimen is permanently deposited and available for researchers at a public repository, the Beipiao Pterosaur Museum of China, Beipiao, Liaoning Province, China (Fig. 1).

**Type Locality and Horizon.** Huanghuatan village, Dapingfang town, Chaoyang City, Liaoning Province, China (Fig. 2); lower part of the Jiufotang Formation, Early Cretaceous (Aptian).

**Differential diagnosis.** The new species is diagnosed based on the following features: quadrate inclined at 150°; medial process of the pterygoid broad and plate-like; dorsal median sulcus of the mandibular symphysis extending up to the first pair of mandibular teeth; dorsally directed odontoid (pseudotooth) of the mandibular symphysis, lacking a

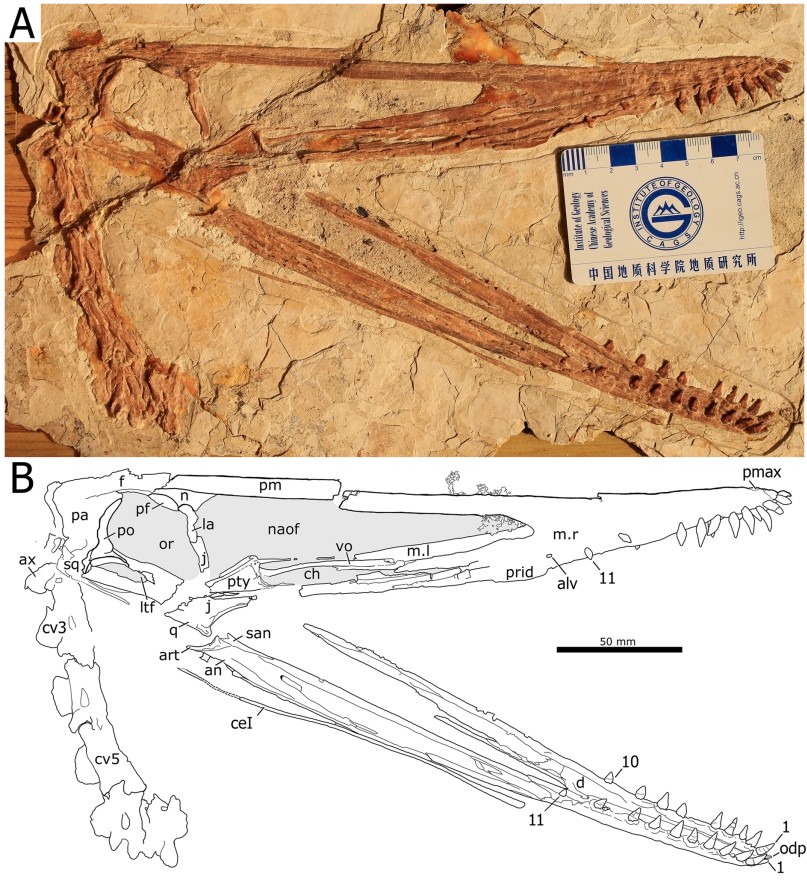

**Figure 1 *Nurhachius luei* sp. nov., BPMC-0204, holotype, photograph, and line drawing.** The scale bar in the line drawing equals 50 mm. Abbreviations: alv, alveolus; an, angular; art, articular; ax, axis; ceI, ceratobranchial I; ch, choana; cv, cervical vertebra; d, dentary; f, frontal; j, jugal; la, lacrimal; m, maxilla; n, nasal; naof, nasoantorbital fenestra; odp, odontoid process; or, orbit; pa, parietal; pf, prefrontal; pmax, premaxilla; po, postorbital; prid, palatal ridge; pty, pterygoid; q, quadrate; vo, vomer. Isolated numbers indicate tooth positions. Note: the visible region of the pterygoid corresponds to the medial process of the bone. Photo by Xuanyu Zhou. Drawing by Maria Eduarda Leal.

foramen on the lateral side and with a blunt occlusal surface; ceratobranchial I of the hyoids accounting for 60% of mandibular length; mandibular teeth extending distally beyond the symphysis.

## Description

**Skull and mandible.** The skull is exposed in right lateral view, with some palatal elements that are visible in dorsal view. The mandible is exposed in right dorsolateral view. The skull is 300 mm long from the squamosal to the premaxillary tip (total skull length), and 74 mm high at its greatest height, which is at the level of the occiput. The nasoantorbital fenestra is long, corresponding to 45% of the total skull length (premaxilla to squamosal) and 55% of the length from the craniomandibular joint to the premaxilla. Anterior to the nasoantorbital fenestra, the long axis of the rostrum is slightly deflected dorsally, as in other istiodactylids (*Wang et al., 2005*; *Andres & Ji, 2006*; *Lü, Xu & Ji, 2008*;

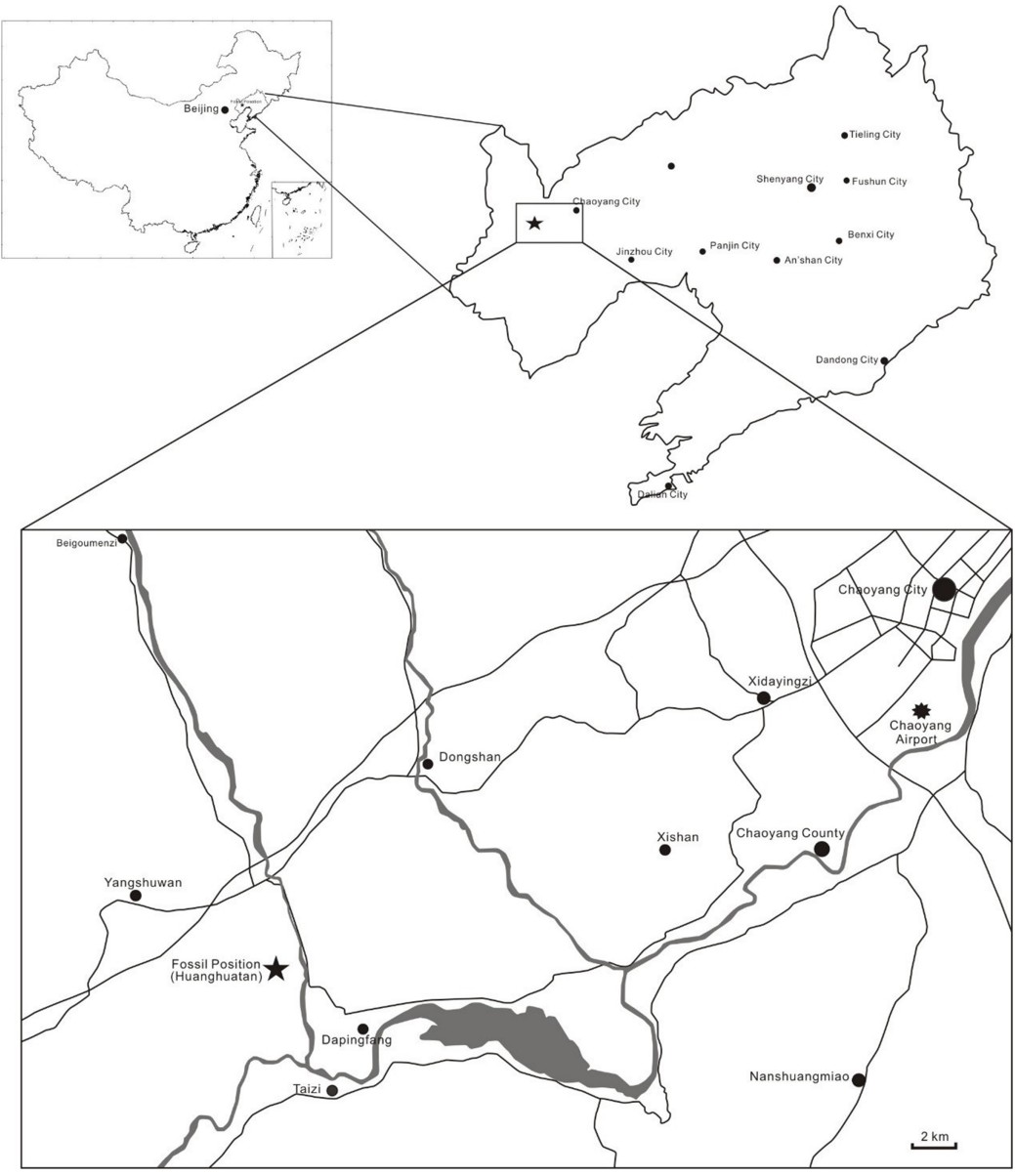

**Figure 2 Location of the site where BPMC-0204 was found.**

*Witton, 2012*), as well as *Ikrandraco avatar* and anhanguerians (*Kellner & Tomida, 2000*; *Wang et al., 2014, 2015*; *Holgado et al., 2019*), but unlike boreopterids (*Lü & Ji, 2005*; *Lü, 2010*; *Jiang et al., 2014*). There is a strong palatal keel extending from pre-narial part of the rostrum to the anterior third of the nasoantorbital fenestra. The craniomandibular joint levels with the anterior margin of the orbit, similarly to both specimens of *N. ignaciobritoi* (see Fig. 3; both specimens, LPM 00023 and IVPP V-13288; see *Wang et al., 2005, 2006*; *Lü, Xu & Ji, 2008*), *Anhanguera* spp. (see *Kellner & Tomida, 2000*) and *Linlongopterus jennyae* (see *Rodrigues et al., 2015*), but unlike *Istiodactylus* spp., in which the joint is located anterior to the orbit (see *Andres & Ji, 2006*; *Witton, 2012*), and

*Ikrandraco avatar* (see *Wang et al., 2015*), *Hamipterus tianshanensis* (see *Wang et al., 2014*) and *Ludodactylus sibbicki* (*Frey, Martill & Buchy, 2003*), in which the joint is located under the middle of the orbit. The orbit is piriform, with the narrowest part being ventral, and without a suborbital vacuity. This is similar to the condition seen in the referred specimen of *N. ignaciobritoi* and unlike the rounded orbit of *Istiodactylus*, which has also a suborbital vacuity (see *Andres & Ji, 2006*; *Lü, Xu & Ji, 2008*; *Witton, 2012*). The infratemporal fenestra is elliptical and much smaller than the orbit. The supratemporal fenestra is poorly preserved.

**Premaxilla and Maxilla.** The premaxilla is fused with the maxilla and the suture is obliterated, thus the boundary between the two bones cannot be traced. Consequently, the premaxillary and maxillary teeth count is unknown. There is no premaxillary crest, as in all other istiodactylids and *Haopterus gracilis*. The rostral tip of the premaxilla exhibits a slight dorsal deflection of palatal anterior tip (Fig. 4), as evidenced from the uplifted positions of the two anteriormost teeth. This is similar to what has already been reported for *Cimoliopterus* and anhanguerians (see *Rodrigues & Kellner, 2013*).

**Nasal and Lacrimal.** The nasal and lacrimal form the anterodorsal margin of the orbit and the posterodorsal margin of the nasoantorbital fenestra. The anterior end of the nasolacrimal coincides with the highest point of the nasoantorbital fenestra, as in both specimens of *N. ignaciobritoi* and also *Ikrandraco avatar* (*Wang et al., 2005*, *2006*, *2015*; *Andres & Ji, 2006*; *Lü, Xu & Ji, 2008*), but unlike *Istiodactylus latidens* and most anhanguerians (e.g., *Anhanguera*, *Tropeognathus*, and *Hamipterus*), except for *Ludodactylus sibbicki*, in which the highest point is posterior to the anterior end of the nasolacrimal (*Campos & Kellner, 1985*; *Wellnhofer, 1987*; *Kellner & Tomida, 2000*; *Wang et al., 2014*; *Frey, Martill & Buchy, 2003*). A nasal descending process cannot be seen in BPMC-0204, possibly because it is still covered by rock. There are no traces of an orbital process of the lacrimal invading the orbit, but the posterior margin of the lacrimal is slightly damaged and a small process similar to the one seen in the holotype of *N. ignaciobritoi* may had been present and got lost (see *Wang et al., 2005*, *2006*; *Andres & Ji, 2006*; *Lü, Xu & Ji, 2008*). The lacrimal contacts the lacrimal process of the jugal at about the mid-height of the posterior margin of the nasoantorbital fenestra. The nasal is bordered dorsally by the premaxilla and by the prefrontal posteroventrally.

**Jugal and Quadratojugal.** The jugal is partially preserved, missing part of the maxillary process and the base of the lacrimal process. The jugal sends a postorbital process to contact the postorbital, separating the orbit and the infratemporal fenestra. Posteriorly, the jugal contacts the quadratojugal, which forms the anteroventral of the infratemporal fenestra.

**Quadrate.** Sutures at the lateral surface of the quadrate are unclear. It is unclear whether the articulation with the mandible is helical or not. The mid-region of the quadrate is lost. The dorsal portion of the quadrate contacts the quadratojugal anteriorly and the squamosal dorsally. The quadrate is inclined backwards at an angle of 150°, unlike both

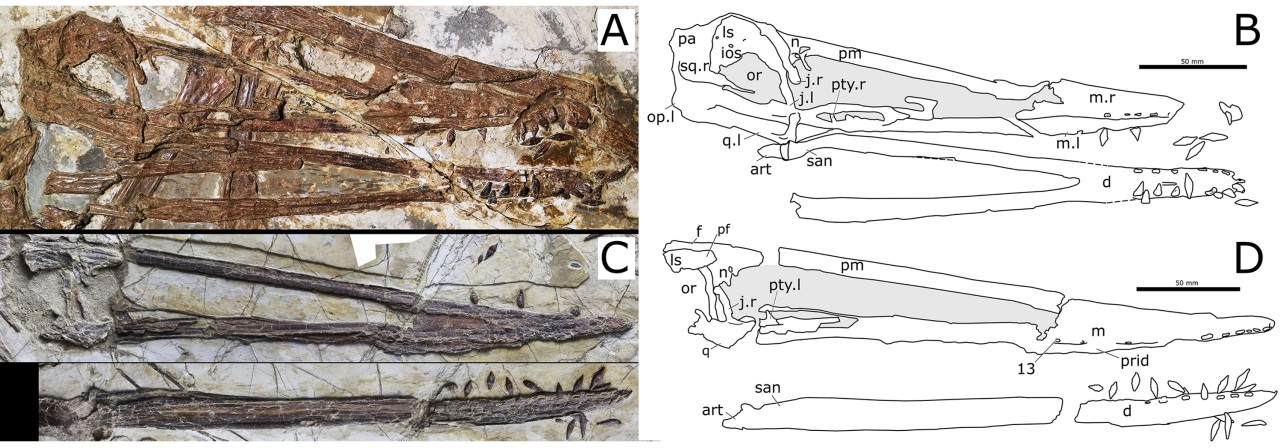

**Figure 3 *Nurhachius ignaciobritoi* specimens, photographs and line drawings.** (A) LPM 00023, referred specimen (former holotype of *"Long-chengpterus zhaoi"*), skull and mandible in right lateral view; and (B) interpretative line drawing. (C) IVPP V-13288, holotype, skull (mirrored), and mandible in right lateral view; and (D) interpretative line drawing. Scale bars equal 50 mm. Abbreviations: art, articular; ch, choana; d, dentary; f, frontal; ios, interorbital septum; j, jugal; ls, laterosphenoid; m, maxilla; n, nasal; op, opisthotic; or, orbit; pa, parietal; pf, prefrontal; pm, premaxilla; prid, palatal ridge; pty, pterygoid; q, quadrate; san, surangular; sq, squamosal. Photographs by Xuanyu Zhou. Drawings by Rodrigo V. Pêgas.

specimens of *N. ignaciobritoi* (Fig. 3), in which it slopes at ~160° (160°4′ in the holotype; *Wang et al., 2005*; 163° in the referred specimen; *Wang et al., 2006*).

**Prefrontal.** The prefrontal is a small bone that forms the anterodorsal margin of the orbit, contacting the nasolacrimal. A suture between these two bones can be seen anteroventrally. The dorsoposterior tip of this bone contacts the frontal.

**Frontal.** The frontal seems to be fused with the premaxilla and parietal, with no visible sutures. It is unclear whether the posterodorsal extension of the frontal forms a blunt and low frontoparietal crest as in *Anhanguera* (see *Kellner & Tomida, 2000*) or not.

**Parietal and Squamosal.** The parietal and squamosal are poorly preserved, especially the latter. The squamosal outline cannot be properly identified. The parietal preserves a shallow depression in its surface that corresponds to the medial wall of the supratemporal fenestra. The dorsal limits of this fossa level with the orbit and extend ventrally to the region of contact between the squamosal and the postorbital.

**Postorbital.** The postorbital is slender and does not exhibit a triangular shape, oppositely to the triangular condition that is seen in anhanguerids (*Kellner & Tomida, 2000*). Instead, it is like a three-pointed star (= concave equilateral hexagon) as in *Haopterus gracilis* (see *Wang & Lü, 2001*); that is, is essentially composed of three connected processes. The anterior region of the postorbital which is formed by the frontal and jugal processes, is arched and forms the posterior margin of the orbit. The squamosal process is shorter than the other processes and separates the supra and infratemporal fenestrae. There is no orbital process of the postorbital invading the orbit, unlike *Istiodactylus* (*Andres & Ji, 2006*; *Witton, 2012*).

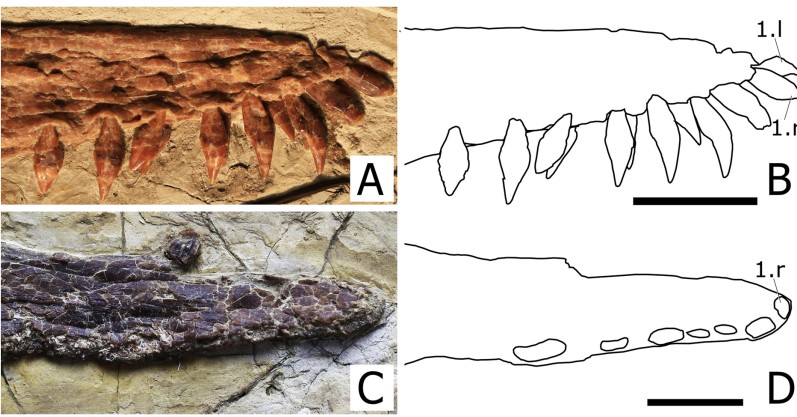

**Figure 4** **Close view of the rostral tip of *Nurhachius* species in right lateral view.** (A) *Nurhachius luei* sp. nov., holotype and (C) *Nurhachius ignaciobritoi*, IVPP V-13288, holotype, mirrored. (B) and (D), respective schematic drawings of (A) and (C), showing the slight dorsal deflection of the palate (notice the positions of the first and second alveoli in both specimens). Numbers indicate tooth positions. Scale bars equal 20 mm. Photos by Xuanyu Zhou. Drawings by Rodrigo V. Pêgas.

**Palatal elements.** Due to crushing, some palatal elements are visible in dorsal view, though few details can be observed. The vomers form a long, slender bony bar that separates the choanae, as in *Hongshanopterus lacustris* (see *Wang et al., 2008a*). Of the pterygoid, only the medial process can be seen. It is large and plate-like as that of *Hongshanopterus lacustris* (see *Wang et al., 2008a*) and, to a lesser extent, the anhanguerids, in which the process is also broad but less medially expanded (*Campos & Kellner, 1985*; *Frey, Martill & Buchy, 2003*). This differs from the slender medial processes of the pterygoid of azhdarchoids (*Pinheiro & Schultz, 2012*; *Kellner, 2013*; *Pêgas, Costa & Kellner, 2018*) or those of the referred specimen of *N. ignaciobritoi* (see *Wang et al., 2006*; *Lü, Xu & Ji, 2008*) and *Ikrandraco avatar* (see *Wang et al., 2015*).

**Dentary.** The dentaries are fused rostrally forming a symphysis that accounts for 36% of total mandibular length, which is 240 mm long. The dorsal surface of the symphysis presents a deep and broad median sulcus that extends anteriorly up to the level of the first pair of teeth (Fig. 5). The rostral tip of the dentary symphysis has an odontoid (pseudotooth), that is located between the first pair of teeth, is smaller than the adjacent tooth crowns and is dorsally directed. The odontoid lacks the neurovascular foramen piercing its surface in the referred specimen of *N. ignaciobritoi* (see *Wang et al., 2006*). The odontoid has the same orientation as that of *Istiodactylus latidens* (see *Witton, 2012*; *Martill, 2014*) and *Lonchodraco giganteus* (see *Rodrigues & Kellner, 2013*, fig. 4E-F), unlike the sub-horizontal odontoids of both specimens of *N. ignaciobritoi* (see *Wang et al., 2005*, *2006*) and *Ikrandraco avatar* (see *Wang et al., 2015*). The symphysis presents 11 tooth positions per side.

**Surangular, Articular, and Angular.** The lateral surface of the posterior region of the right mandibular ramus is composed by the surangular, angular, and articular. A suture separates the long anterior process of the surangular from the dentary dorsally. Posteriorly,

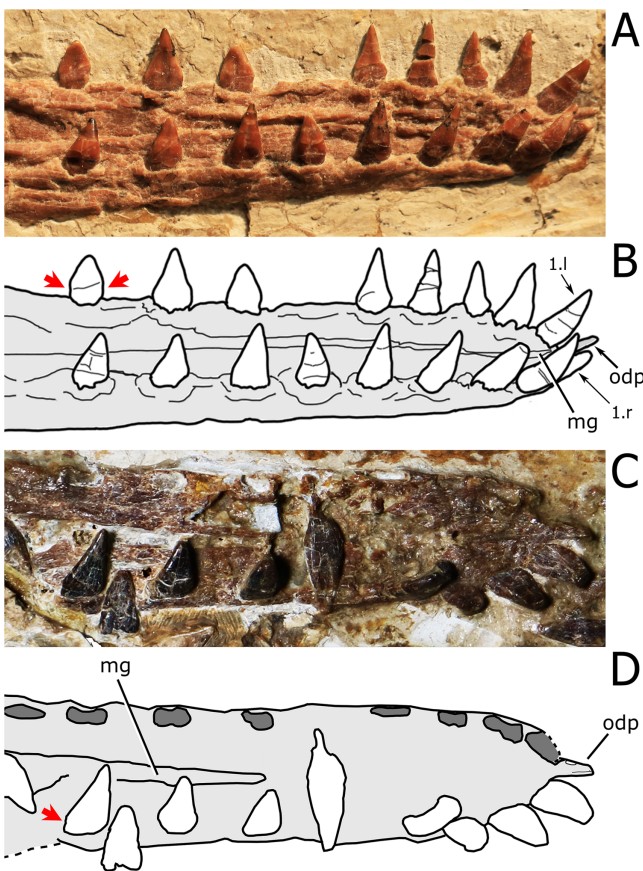

**Figure 5 Close view of the dentary symphysis of *Nurhachius* species.** (A) *Nurhachius luei* sp. nov. holotype in dorsolateral view, and (B) line drawing. (C) *Nurhachius ignaciobritoi*, LPM 00023, referred specimen, occlusal view, and (D), line drawing. Abbreviations: mg, median groove; odp, odontoid process. Numbers indicate tooth positions. Red arrows indicate the mesiodistal constriction between crown and root. Photos by Xuanyu Zhou. Drawings by Maria Eduarda Leal and Rodrigo V. Pêgas.

the surangular becomes deeper and is sutured with the angular. The boundary between the angular and the dentary, however, cannot be distinguished, nor the boundary between the angular and the articular. The articular forms the posterior part of the mandible, including the articular surface for the quadrate and the retroarticular process, which is pointed, dorsoventrally low and distally tapering.

**Hyoid.** Only the right ceratobranchial I is exposed along the ventral margin of the right mandibular ramus (Fig. 1). A small portion of the posterior part is missing. The ceratobranchial I is a long rod-like bone extending along the whole length of the mandibular rami.

**Dentition.** There are 12 tooth positions along the right side of the upper jaw and 11 tooth positions along each side of the lower jaw, with an inferred total count of 46 tooth positions. The first two teeth of the upper jaw (which are presumably premaxillary teeth) are procumbent. The first tooth forms an angle of 130° with the main axis of the rostrum,

while the second forms an angle of 123°. The third tooth is also slightly procumbent, forming an angle of 100° with the palatal plane. All subsequent teeth are perpendicular to the main axis of the rostrum. The first two dentary teeth are also slightly procumbent. The last two alveoli of the right maxilla are empty, and the last one is placed just anterior to the level of the rostral end of the nasoantorbital fenestra. All of the crowns are triangular in labiolingual view and labiolingually compressed, as typical of the Istiodactylidae. The base of the crowns is mesiodistally inflated. The lingual surface of the crown is concave with a well-marked basoapical depression and a low transversal convexity at the base, that forms a lingual cingulum. The labial surface is mostly convex with a shallow concavity in the middle of the basal part of the crown. No carinae are present along the mesial and distal cutting margins of the crowns. The same features occur in the crowns of the holotype of *N. ignaciobritoi*. The teeth exhibit a slight constriction between crown and root (Fig. 5). This feature is also shared with *N. ignaciobritoi* (see *Wang et al., 2005*).

The first nine pairs of teeth of the upper jaw are large and subequal in size. Their apicobasal total length of their crowns is about 12 mm and the crown apicobasal length is about seven mm, and the mesiodistal width of the socket is four mm. The smallest crown is six mm in apicobasal length and two mm in mesiodistal width.

*Nurhachius luei* sp. nov. also presents an interesting pattern of tooth replacement. Two teeth occur in the 10th alveolus of the right dentary: a large functional one and a not yet fully erupted replacement tooth. The replacement tooth was erupting anterolabially to the functional tooth, instead of posterolingually as reported in other pterodactyloid pterosaurs like *Anhanguera* (see *Kellner & Tomida, 2000*; *Fastnacht, 2001*) and "*Cearadactylus*" *ligabuei* (see *Dalla Vecchia, 1993*).

**Cervical vertebrae.** Seven cervical vertebrae are preserved, including the atlas-axis complex (although the atlas itself cannot be identified). They are articulated, except for the seventh vertebra, which is disarticulated but still contacting the sixth vertebra. The third cervical, with 38.4 mm, is longer than the fourth through seventh cervicals which are of similar length. The neural spine is damaged in most cervicals, except that of the fourth vertebra, which is high and with a peculiar shape (its anterior margin is anteriorly inclined). The apex of the neural spine is gently rounded. The postzygapophyses are posterodorsally oriented. The centrum extends posterior to the postzygapophyses. In the third to the seventh cervicals, a large pneumatic foramen can be seen on the posterior half of the centrum below the neural arch.

## Phylogenetic analysis results

The phylogenetic analysis by parsimony produced 51 most parsimonious trees with a minimum length of 360 steps, with minimum consistency index of 0.642 and retention index of 0.862. As suspected by *Witton (2012)*, *Hongshanopterus lacustris* and *Haopterus gracilis* were found to be closely related to the Istiodactylidae in the strict consensus tree (Fig. 6), but they fall outside them. The Istiodactylidae have been defined by *Andres, Clark & Xu (2014)* as the least inclusive clade containing *Istiodactylus latidens* and *N. ignaciobritoi* Wang et al. 2005. In the strict consensus tree (Fig. 6), the Istiodactylidae

contain *Nurhachius*, *Liaoxipterus brachyognathus* and *Istiodactylus*. *N. ignaciobritoi* and *N. luei* sp. nov. were recovered as sister taxa. *Istiodactylus latidens* and *Istiodactylus sinensis* were also recovered as sister taxa and *Liaoxipterus brachyognathus* is the sister-taxon of *Istiodactylus*.

*Istiodactylus* has the following five synapomorphies: presence of a suborbital opening (character 11, state 1); prenarial portion of the rostrum less than 20% the skull length (character 25, state 0); presence of an orbital process in the jugal (character 56, state 1); and sharp carinae in the teeth (character 96, state 1). *Istiodactylus* shares with *Liaoxipterus brachyognathus* the following synapomorphies: subparallel lateral margins of the jaws (character 24, state 1); mandibular symphysis shorter than 33% of the mandible length (character 78, state 1); and rounded outline of the rostral end of the mandible (character 79, state 0).

The genus *Nurhachius* is characterized by the following five synapomorphies: piriform orbit (character 7, state 2); cranio-mandibular articulation under the anterior margin of orbit (character 58, state 2); dorsal deflection of the palatal anterior tip (character 71, state 1); teeth crowns with labial and lingual depressions (character 100, state 1); teeth with a mesiodistal constriction between crown and root (character 101, state 1). The presence of a dorsal deflection of the palatal anterior tip represents a homoplasy with Anhangueria + *Cimoliopterus*.

The Istiodactylidae share the following seven synapomorphies: ventral margin of the nasoantorbital fenestra longer than 40% of skull length (character 4, state 1); orbit reaching high in the skull, with the dorsal margin surpassing the dorsal margin of the nasoantorbital fenestra (character 10, state 1); skull height (exclusive of cranial crests) over 25% of the jaw length (character 23, state 1); lacrimal process of jugal inclined posteriorly (character 54, state 2); helical jaw-joint absent (character 59, state 0); palatal occlusal surface: strong palatal ridge confined to the posterior portion of the palate (character 71, state 3); teeth confined to about the anterior third of the jaws (character 86, state 3).

*Hongshanopterus lacustris* results to be the sister-taxon of the Istiodactylidae. The clade *Hongshanopterus lacustris* + Istiodactylidae presents two synapomorphies: tooth crowns strongly compressed laterally (character 95, state 2); and mesial crowns under twice as long as wide (character 97, state 0).

*Haopterus gracilis* results as the sister-taxon of *Hongshanopterus lacustris* + Istiodactylidae, sharing teeth that are confined to the anterior half of the jaws (character 86, state 2).

The Istiodactylidae, *Hongshanopterus lacustris* and *Haopterus gracilis* share with *Ikrandraco avatar* four synapomorphies: narrow lacrimal process of jugal (character 53, state 1); quadrate inclination relative to the ventral margin of the skull is 150° or more (character 57, state 3); tooth crowns slightly compressed laterally (character 95, state 2); lingual cingulum present at the base of tooth crown (character 102, state 1). The last two character states are also shared with *Lonchodraco giganteus*.

Bremer support values were 1 for the genus *Istiodactylus*, 3 for *Istiodactylus* + *Liaoxipterus*, 5 for the genus *Nurhachius*, 3 for Istiodactylidae, 3 for *Hongshanopterus* +

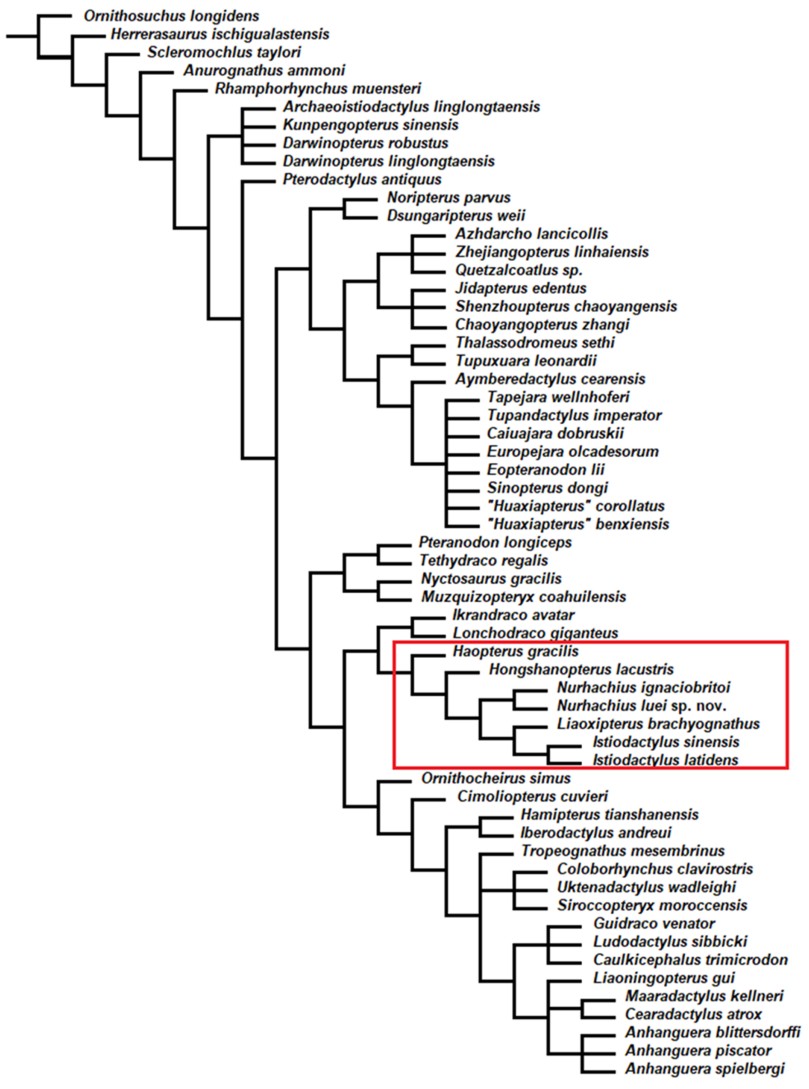

**Figure 6 *Nurhachius luei* sp. nov. phylogenetic relationships.** Strict consensus tree of 51 most parsimonious trees. Tree length is 360, consistency index 0.642 and retention index 0.864. The red rectangle indicates the Istiodactylidae and its two closest taxa.

Istiodactylidae, 1 for *Haopterus* + (*Hongshanopterus* + Istiodactylidae), 1 for *Ikrandraco* + *Lonchodraco* and 2 for the clade that joins all of these taxa.

# DISCUSSION

For over a century, *Istiodactylus latidens* was the only known istiodactylid (*Witton, 2012*). In the last 15 years, three new istiodactylids have been reported from the Jiufotang Formation of China: *N. ignaciobritoi* (described in 2005); *Istiodactylus sinensis* and *Longchengpterus zhaoi* (both described in 2006); and *Liaoxipterus brachyognathus* (originally described in 2005 as a purported ctenochasmatid and referred to the Istiodactylidae in 2008; *Zhiming & Junchang, 2005*; *Wang et al., 2005*, *2006*; *Andres & Ji, 2006*; *Lü, Xu & Ji, 2008*). However, the validity of some of them is debated. According to *Lü, Xu & Ji (2008)*, the holotypes of *Longchengpterus zhaoi* and *N. ignaciobritoi* are

indistinguishable, sharing general skull shape and tooth morphology. Therefore, *Longchengpterus zhaoi* was considered a junior synonym of *N. ignaciobritoi* by *Lü, Xu & Ji (2008)*. *Witton (2012)* provisionally considered both of them as valid and distinct taxa, coding them separately in his phylogenetic analysis though without discussing it further. They were coded differently as for tooth count and spacing, with *Nurhachius* that was considered to have more numerous and more spaced teeth. However, both specimens exhibit a similar number of upper teeth (13 pairs in the holotype of *N. ignaciobritoi* and 12 pairs in Longchengpterus *zhaoi*) and similar spacing (*Wang et al., 2005*; *Lü, Xu & Ji, 2008*). Furthermore, the holotypes and only specimens of *Longchengpterus zhaoi* and *N. ignaciobritoi* share further features that are unique within istiodactylids (Fig. 3): the high quadrate inclination (~160°), the reduced medial process of the pterygoid, the upper dentition ending at the level of the nasoantorbital fenestra, and the sub-horizontal odontoid in the mandibular symphysis. Therefore, we follow *Lü, Xu & Ji (2008)* in considering *Longchengpterus zhaoi* as a junior synonym of *N. ignaciobritoi*.

*Nurhachius luei* sp. nov. is an istiodactylid based on the following features: nasoantorbital fenestra longer than 40% of skull length, dentary symphysis less than 33% of mandible length; and triangular, labiolingually compressed tooth crowns. It shares with *N. ignaciobritoi* a piriform orbit, a dorsally deflected palatal anterior tip, a cranio-mandibular articulation positioned under the anterior margin of the orbit (Fig. 3), tooth crowns with labial and lingual depressions, and teeth with mesiodistal constrictions between crown and root (Figs. 5 and 6). All these features are present in the holotype of *N. ignaciobritoi*, while in the referred specimen the dorsal deflection of the palatal anterior tip cannot be assessed as the rostrum tip is missing. Concerning the crown, with both a labial and a lingual depression, we note that this morphology is reflected in the shape of the alveoli in *N. ignaciobritoi*, with concave labial and lingual margins (Fig. 5). The dorsal deflection of the palatal anterior tip is present in the rostral tip of the skull of the holotype of *N. ignaciobritoi*, as evidenced from the dorsal position of the first pair of alveoli (see Figs. 4C and 4D), although it was not mentioned in the original description (*Wang et al., 2005*), as well as in the holotype of *N. luei*. This character was utilized in a data matrix for the first time by *Rodrigues & Kellner (2013)*, and resulted to be a synapomorphy of Anhangueria + *Cimoliopterus*. According to our phylogenetic analysis (Fig. 7), this feature was independently acquired by *Nurhachius* and Anhangueria + *Cimoliopterus*.

*Nurhachius luei* sp. nov. differs from *N. ignaciobritoi* in the following features: the quadrate is inclined at 150° instead of the ~160° of *N. ignaciobritoi*; the medial process of the pterygoid is broad and plate-like, whereas it is reduced in *N. ignaciobritoi* (see Fig. 3); the dorsal median sulcus of the mandibular symphysis extends up to the first pair of teeth, whereas it reaches the sixth pair of teeth in *N. ignaciobritoi* (see Fig. 5); the odontoid (pseudotooth) lacks a lateral foramen, whereas a foramen is present in the referred specimen of *N. ignaciobritoi* (see *Martill, 2014*, fig. 7C-D); the odontoid has a blunt occlusal surface, whereas the surface is sharp in *N. ignaciobritoi* (Fig. 5); the odontoid is dorsally directed, whereas it is anterodorsally directed in *N. ignaciobritoi* (but see *Martill,*

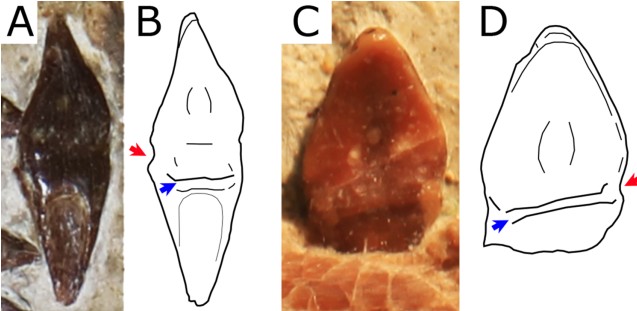

**Figure 7 Close view of the dentition in *Nurhachius* species.** (A) *Nurhachius ignaciobritoi*, LPM 00023, referred specimen, isolated tooth in lingual view, and (B) line drawing. (C) *Nurhachius luei* sp. nov. holotype, ninth left mandibular tooth in lingual view, and (D) line drawing. Red arrows indicate the mesiodistal constrictions between crown and root. Blue arrows indicate the horizontal elevation at the base of the crown (cingulum). Photos by Xuanyu Zhou. Drawings by Rodrigo V. Pêgas.

*2014*, p. 57, right column, lines 21–23); and the ceratobranchial I of the hyoid apparatus accounts for 60% of the mandibular length, whereas it accounts for 35% of the mandibular length in *N. ignaciobritoi* (Fig. 5).

Concerning quadrate inclination, we do not regard this variation as intraspecific as the variation does not surpass 3° in the archaeopterodactyloid *Pterodactylus antiquus* (specimens BSPG AS I 739, BSPG 1929 I 18, BMMS 7; see *Bennett, 2013*; *Vidovic & Martill, 2014*), 3° in the anhanguerian *Hamipterus tianshanensis* (IVPP V18931.1, holotype, and IVPP V18935.1, paratype; see *Wang et al., 2014*), 5° in *Aerodactylus scolopaciceps* (BSPG 1883 XVI 1, BSPG 1937 I 18, BSPG AS V 29 a/b; see *Vidovic & Martill, 2014*), and is less than 6° in *Pteranodon longiceps* (specimens YPM 1177, USNM 13656, KUVP 2212, KUVP 27821) and also in *Pteranodon sternbergi* (specimens FHSM VP 339, YPM 1179, UALVP 24238; see *Bennett, 1994, 2001*). Quadrate inclination has indeed been regarded as diagnostic before for tapejarids (*Kellner, 2013*) and *Pteranodon* (*Bennett, 1994*). We regard this feature as potentially diagnostic at least until further specimens of each species of *Nurhachius* are found.

Both specimens of *N. ignaciobritoi* come from the upper part of the Jiufotang Formation (see *Wu et al., 2018*), while the holotype of *N. luei* comes from the lowermost part of the Jiufotang Formation. This stratigraphic distribution might be suggestive of an anagenetic link between the two species, similar to the case of *Pteranodon longiceps* (from the upper Smoky Hill Chalk) and *Pteranodon sternbergi* (from the lower Smoky Hill Chalk) according to *Bennett (1994)*, but see taxonomic controversies (*Kellner, 2010, 2017*; *Martin-Silverstone et al., 2017*; *Acorn et al., 2017*). The same has been speculated as a possible explanation for the occurrence of multiple species of *Anhanguera* (*Pinheiro & Rodrigues, 2017*), *Thalassodromeus* and *Tupuxuara* (*Pêgas, Costa & Kellner, 2018*) in the Romualdo Formation. However, these cases still lack stratigraphic control for support, and the time resolution of the Romualdo Formation is still in question (see *Pinheiro & Rodrigues, 2017*).

*Wang et al. (2008a)* were unable to differentiate *Liaoxipterus brachyognathus* from *Longchengpterus zhaoi*, and suggested that *Longchengpterus zhaoi* could be a junior

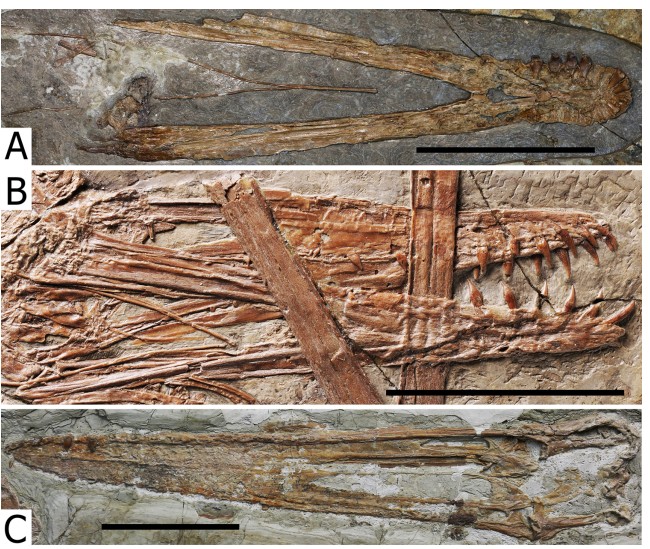

**Figure 8** **Other istiodactylids and close taxa.** (A) *Liaoxipterus brachyognathus*, CAR-0018, holotype, lower jaw in dorsal view. (B) *Haopterus gracilis*, IVPP V11726, holotype, skull in right lateral view. (C) *Hongshanopterus lacustris*, IVPP V14582, holotype, skull in ventral view. All scale bars equal 50 mm. (A) and (C) by Xuanyu Zhou; (B) by Shunxing Jiang (courtesy of IVPP).

synonym of *Liaoxipterus brachyognathus*. However, as observed by *Lü, Xu & Ji (2008)*, the rostral end of the mandibular symphysis is rounded in *Liaoxipterus brachyognathus* (in dorsal view, as it is in *Istiodactylus latidens*), whereas it is triangular in "*Longchengpterus zhaoi*." Furthermore, as coded by *Andres, Clark & Xu (2014)*, both jaws show an attenuated taper (in occlusal view) in *Longchengpterus zhaoi*, while the lateral margins of the lower jaw are sub-parallel in *Liaoxipterus brachyognathus* (Fig. 8A) as in *Istiodactylus latidens*. Additionally, the mandibular symphysis of *Liaoxipterus brachyognathus* is relatively stouter than that of *Longchengpterus zhaoi*: their length/width ratios are 0.43 and 0.27, respectively. It is worthy of being noticed that the mandibular symphysis of *Longchengpterus zhaoi* is incorrectly drawn in *Martill (2014*, fig. 7B*)*, as resulting to be much shorter than it is. The actual configuration can be clearly assessed in the description by *Lü, Xu & Ji (2008)* and in Fig. 5A. We thus follow *Lü, Xu & Ji (2008)* and *Witton (2012)* in considering *Liaoxipterus brachyognathus* as distinct from *Longchengpterus zhaoi*, which we consider as a junior synonym of *N. ignaciobritoi*.

*Lü, Xu & Ji (2008)* and *Witton (2012)* noticed that comparison between *Liaoxipterus brachyognathus* and *Istiodactylus sinensis* is very limited because the former is represented by a mandible exposed in occlusal view, while the latter is a partial skeleton including a mandible exposed in lateral view. However, *Liaoxipterus brachyognathus* differs from *Istiodactylus* in the lack of mesial carinae, according to *Lü, Xu & Ji (2008)* and the dataset of *Andres, Clark & Xu (2014)*. We thus follow these authors in considering *Liaoxipterus brachyognathus* as a valid taxon.

According to our phylogenetic analysis, *Istiodactylus* is monophyletic, comprising *Istiodactylus* latidens and *Istiodactylus* sinensis. *Liaoxipterus brachyognathus* is the sister-taxon of *Istiodactylus*, and *Nurhachius* is the sister-taxon of *Liaoxipterus*

*brachyognathus* + *Istiodactylus*, in agreement with the results of the phylogenetic hypothesis published by *Longrich, Martill & Andres (2018)*. In our analysis, *N. luei* results to be the sister-taxon of *N. ignaciobritoi*, supporting their congeneric status. The relationships within the Istiodactylidae obtained in our analysis are similar to those found by *Andres, Clark & Xu (2014)*, but *Longchengpterus zhaoi* is not the sister-taxon of *N. ignaciobritoi* in the cladogram of figure S2 of *Andres, Clark & Xu (2014)*.

*Haopterus gracilis* was first described by *Wang & Lü (2001)* and referred to the Pterodactylidae. However, it resulted to be close to *Istiodactylus latidens* in the 50% majority-rule tree by *Lü, Xu & Ji (2008)* and formed a polytomy with *Nurhachius* and *Istiodactylus* in the strict consensus tree by *Lü et al. (2009)*. *Hongshanopterus lacustris* was described by *Wang et al. (2008a)* and interpreted as a primitive istiodactylid. In the strict consensus tree by *Witton (2012)*, the Istiodactylidae include *N. igniaciobritoi*, *Longchengpterus zhaoi*, *Istiodactylus latidens*, *Istiodactylus sinensis*, and *Liaoxipterus brachyognathus*. *Haopterus gracilis*, and *Hongshanopterus lacustris* form a polytomy with *Pteranodon longiceps*, *Coloborhynchus spielbergi*, and the Istiodactylidae (*Witton, 2012*).

In the phylogenetic analysis of *Andres, Clark & Xu (2014*; fig. S2*)*, *Haopterus gracilis* results to be a basal eupterodactyloidean and *Hongshanopterus* a basal ornithocheiromorph. Recently, *Holgado et al. (2019)* have published a phylogenetic hypothesis in which *Hongshanopterus lacustris* would be the sister-group of the Istiodactylidae (although it is erroneously reported within this clade as the basal member in their fig. 5A), while *Haopterus gracilis* would be closer to anhanguerians than to istiodactylids (see *Holgado et al., 2019*).

In our analysis, *Haopterus gracilis* and *Hongshanopterus lacustris* are closely related to the Istiodactylidae, as found by *Lü, Xu & Ji (2008)* and *Wang et al. (2008a)*, respectively. *Hongshanopterus lacustris* results to be the sister-taxon of the Istiodactylidae, as in the analysis by *Holgado et al. (2019)*. *Hongshanopterus lacustris* shares with the istiodactylids the presence of labiolingually compressed teeth with triangular crowns. *Haopterus gracilis* results to be the sister-taxon of *Hongshanopterus lacustris* + Istiodactylidae, a relationship that is supported by the possession of a dentition restricted to the anterior half of the jaws.

*Haopterus*, *Hongshanopterus*, and istiodactylids share also the presence of a lingual cingulum in the tooth crown, a feature that occurs also in *Ikrandraco avatar*. A lingual cingulum can be seen in *N. luei* and *N. ignaciobritoi* (Fig. 6). The same feature has been previously reported for *Liaoxipterus brachyognathus* (see *Lü, Xu & Ji, 2008*) and depicted for *Ikrandraco avatar* (see the second figure of *Wang et al., 2015*). In *Haopterus gracilis*, the labiodistal view of the third right upper tooth presents a lingually oriented convexity that also suggests the presence of this feature (Fig. 8).

*Ikrandraco avatar* shares with istiodactylids also a narrow lacrimal process of the jugal and a quadrate inclined at 150° or over (the inclination of the quadrate is unknown in *Haopterus gracilis* and *Hongshanopterus lacustris*). *Ikrandraco avatar* and *Haopterus gracilis* also exhibit a certain degree of labiolingual compression of the teeth, at least in the distal part of the dentition (*Wang & Lü, 2001*; *Wang et al., 2015*), though not to the same degree seen in the istiodactylids and *Hongshanopterus*. The last two mandibular alveoli

preserved in the holotype of *Lonchodraco giganteus* (the sister-taxon of *Ikrandraco avatar* in our analysis) are also labiolingually narrow (see *Martill, 2011*; *Rodrigues & Kellner, 2013*).

Furthermore, *Ikrandraco avatar* and *Lonchodraco giganteus* also share with istiodactylids the presence of an odontoid, which is anterodorsally oriented in the former and dorsally oriented in the latter (see *Rodrigues & Kellner, 2013*; *Wang et al., 2015*).

A close relationship among *Ikrandraco avatar*, *Lonchodraco giganteus* and istiodactylids is found here for the first time. *Ikrandraco avatar* formed a polytomy with the Istiodactylidae, *Cimoliopterus* and the Anhangueria in the phylogenetic analysis by *Wang et al. (2015)* and *Lonchodraco giganteus* is outside the Lanceodontia in the phylogenetic analysis by *Longrich, Martill & Andres (2018)*.

*Archaeoistiodactylus linglongtaensis* is based on the sole holotype (JPM04-0008), including fragments of skull and one displaced maxillary tooth, a partial lower jaw in occlusal view with two teeth in place, an almost complete forelimb, a femur and a tibia. It is from the Middle Jurassic (Bathonian-Oxfordian) Tiaojishan Formation. *Archaeoistiodactylus linglongtaensis* was described by *Lü & Fucha (2010)* who interpreted it as the "ancestor form of the known istiodactylid pterosaur (sic)" (*Lü & Fucha, 2010*, p. 113). *Lü & Fucha (2010)* observed that JPM04-0008 and the istiodactylids share teeth with triangular crowns and an odontoid (pseudotooth) on the mandibular symphysis. That odontoid was mistaken for a mid-line, unpaired tooth by *Sullivan et al. (2014)*, but it had been explicitly described as a bony process by *Lü & Fucha (2010*, p. 116*)*. *Lü & Fucha (2010)* also observed that the single maxillary tooth is recurved as in *Hongshanopterus lacustris*, and reported the presence of a warped deltopectoral crest in the humerus, which is a diagnostic feature of the Pteranodontoidea (*Kellner, 2003*). They noted that *Archaeoistiodactylus linglongtaensis* differs from istiodactylids and all other pterodactyloids in the relatively short fourth metacarpal and in the presence of tibia, and second and third phalanges of the wing digit with subequal lengths. If actually a pterodactyloid, it would represent one of the oldest occurrences of the Pterodactyloidea, being coeval or even older than the Callovian-Oxfordian basalmost pterodactyloid *Kryptodrakon progenitor* (see *Andres, Clark & Xu, 2014*).

Its identification as a pterodactyloid was disputed by *Martill & Etches (2013)*, who affirmed that JPM04-0008 is probably a badly preserved specimen of *Darwinopterus*, though they did not present any evidence to support this statement. According to *Sullivan et al. (2014)*, the short fourth-metacarpal, the long humerus and short first wing phalanx are typical of non-pterodactyloid pterosaurs (see *Kellner, 2003*; *Unwin, 2003*; *Andres, Clark & Xu, 2010*), thus JPM04-0008 is not a pterodactyloid. These features united to the presence of a confluent nasoantorbital fenestra in JPM04-0008, led *Sullivan et al. (2014)* to interpret *Archaeoistiodactylus linglongtaensis* as a basal monofenestratan.

However, *Archaeoistiodactylus linglongtaensis* has never been included in any phylogenetic analysis to test its basal monofenestratan affinity, thus it was included in the analysis performed in this paper. Our results (Fig. 6) confirm the interpretation by *Sullivan et al. (2014)*. *Archaeoistiodactylus linglongtaensis* lacks the following pterodactyloid features: humerus length under 1.5 times metacarpal IV length; ulna under double the length of metacarpal IV; and femur subequal to or shorter than metacarpal IV. The humerus of

JPM04-0008 is crushed and the original orientation of the deltopectoral crest cannot be assessed. Differently from pterodactyloids, the deltopectoral crest of JPM04-0008 is confined to the proximal region of the humerus (*Wang et al., 2009*). *Archaeoistiodactylus linglongtaensis* also lacks pneumatic foramina on the centra of the mid-cervical vertebrae, which is a diagnostic feature of the Dsungaripteroidea (the least inclusive clade containing *Nyctosaurus* and *Quetzalcoatlus*, which includes also the Istiodactylidae; *Kellner, 2003*; *Andres, Clark & Xu, 2014*). Furthermore, *Archaeoistiodactylus linglongtaensis* exhibits low neural spines, like wukongopterids (see *Wang et al., 2009*, *2010*; *Lü et al., 2009*, *2011*; *Cheng et al., 2017*) and unlike istiodactylids (see *Wang et al., 2006*; *Lü, Xu & Ji, 2008*).

The dentition of *Archaeoistiodactylus linglongtaensis* is indeed reminiscent of that of the Istiodactylidae due to the short triangular aspect of the crowns in labiolingual view. However, this feature is also present in the wukongopterids *Wukongopterus lii*, *Darwinopterus robustodens*, *D. linglongtaensis*, and *Kunpengopterus sinensis*, though not in *D. modularis* (see *Wang et al., 2009*, *2010*; *Lü et al., 2009*, *2011*; *Cheng et al., 2017*). Furthermore, in *Archaeoistiodactylus linglongtaensis* the alveoli are circular (*Lü & Fucha, 2010*), as in wukongopterids, not labiolingually compressed triangular teeth as in istiodactylids. The presence or absence of an odontoid in the lower jaw cannot be confidently assessed in *Wukongopterus* and *Darwinopterus*, but can be seen in a specimen referred to *Kunpengopterus sinensis* (see *Cheng et al., 2017*), in convergence with the istiodactylids. Finally, *Archaeoistiodactylus linglongtaensis* shares with *Darwinopterus* and *Kunpengopterus*, but not with *Wukongopterus*, the subequal in length second and third phalanges of the wing digit. Thus, *Archaeoistiodactylus linglongtaensis* may be closely related to *Darwinopterus* or *Kunpengopterus*. In our analysis, *Archaeoistiodactylus linglongtaensis* falls in a polytomy with *D. linglongtaensis, D. robustodens* and *Kunpengopterus sinensis* (Fig. 6). However, we were unable to access the specimen first-hand and further scrutiny is desirable in order to confirm or deny this affinity.

## CONCLUSIONS

The new specimen here described represents the second species for the genus *Nurhachius*, previously restricted to its type-species *N. ignaciobritoi*. A slight dorsal deflection of the palatal anterior tip revealed to be a synapomorphy of *N. ignaciobritoi* and *N. luei*. That feature was previously thought to be restricted to the Anhangueria and *Cimoliopterus*. Unlike other pterodactyloids, the holotype of *N. luei* sp. nov. shows an anterolabial tooth replacement. The position of *Hongshanopterus lacustris* and *Haopterus gracilis* as close taxa to the Istiodactylids is supported by the performed phylogenetic analysis. *Ikrandraco avatar* and *Lonchodraco giganteus* resulted to be sister taxa, and closer to istiodactylids than to other lanceodontians. The phylogenetic analysis supports the reinterpretation of *Archaeoistiodactylus linglongtaensis* as a non-pterodactyloid monofenestratan, probably a wukongopterid.

## ACKNOWLEDGEMENTS

We thank Shu'an Ji and Xuefang Wei (IG-CAGS, Institute of Geology, Chinese Academy of Geological Sciences) for the help all along. Thanks to Cunyu Liu (BPMC, Beipiao

Pterosaur Museum of China), Dongyu Hu (SNU, Shenyang Normal University), Xiaolin Wang & Shunxing Jiang (IVPP, Institute of Vertebrate Paleontology and Paleoanthropology) for access to specimens under their care. RVP thanks Kamila Bandeira, Lucy Souza, and Natan Brilhante (Museu Nacional/UFRJ) for technical help with image software. We thank Zoological Museum (SNM), Copenhagen University for hospitality during X.Y. Zhou's and R.V. Pêgas' stay in Copenhagen with access to important pterosaur specimens, and for M.E.C. Leal's status as guest researcher, and N. Bonde's work space as emeritus (and Senior Scientist, Fur Museum). Thanks to Fabio M. Dalla Vecchia, Felipe Pinheiro, Chris Bennett, and an anonymous reviewer for their thoughtful and constructive critiques; and to editor Graciela Piñeiro for her kind attention.

### Funding

This work was supported by the National Natural Science Foundation of China (Grant No. 41672019, 41688103, 41790452). The funders had no role in study design, data collection and analysis, decision to publish, or preparation of the manuscript.

### Grant Disclosures

The following grant information was disclosed by the authors:
National Natural Science Foundation of China: 41672019, 41688103, 41790452.

### Competing Interests

The authors declare that they have no competing interests.

### Author Contributions

- Xuanyu Zhou conceived and designed the experiments, performed the experiments, analyzed the data, prepared figures and/or tables, authored or reviewed drafts of the paper, approved the final draft.
- Rodrigo V. Pêgas conceived and designed the experiments, performed the experiments, analyzed the data, prepared figures and/or tables, authored or reviewed drafts of the paper, approved the final draft.
- Maria E.C. Leal conceived and designed the experiments, performed the experiments, analyzed the data, prepared figures and/or tables, authored or reviewed drafts of the paper, approved the final draft.
- Niels Bonde conceived and designed the experiments, performed the experiments, analyzed the data, authored or reviewed drafts of the paper, approved the final draft.

### New Species Registration

The following information was supplied regarding the registration of a newly described species:

Publication LSID: urn:lsid:zoobank.org:pub:03EF173E-4AB5-4C74-B80C-A6AAFA65E61C

*Nurhachius luei* sp. nov. LSID: urn:lsid:zoobank.org:act:6F93DC7F-20A7-4CBC-8A38-1D6C802A1906.

## Data Availability

The raw data are available in the Supplemental Files.

The specimens/material are stored in: The Beipiao Pterosaur Museum of China (Chaoyang) under the accession number BPMC-0204: *Nurhachius ignaciobritoi* (both specimens, accession numbers: LPM 00023 and IVPP V-13288), *Liaoxipterus brachyognathus* holotype (CAR-0018), *Hongshanopterus lacustris* holotype (IVPP V14582) and *Haopterus gracilis* holotype (IVPP V11726).

Abbreviations: CAR: Jilin University; IVPP: Institute of Vertebrate Paleontology and Paleoanthropology, Chinese Academy of Sciences; LPM: Liaoning Paleontological Museum at Western Liaoning Institute of Mesozoic Paleontology, Shenyang Normal University; IVPP, Institute of Vertebrate Paleontology and Paleoanthropology, Beijing, China.

## Supplemental Information

Supplemental information for this article can be found online at http://dx.doi.org/10.7717/peerj.7688#supplemental-information.

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
