# Peer review of "Nurhachius luei, a new istiodactylid pterosaur (Pterosauria, Pterodactyloidea) from the Early Cretaceous Jiufotang Formation of Chaoyang City, Liaoning Province (China) and comments on the Istiodactylidae"

_PeerJ, doi:10.7717/peerj.7688_

## Round 0.1 · original submission · Major Revisions

Dear authors,

Thanks for submitting your work for consideration of publication at PeerJ.
We have now three review reports and all of them considered that your manuscript on “A new istiodactylid pterosaur from the Early Cretaceous Jiufotang Formation of Chaoyang City, Liaoning Province; and comments on the group” constitutes an important contribution to the pterosaur pterodactyloid clade.

However, several changes and improvements are needed, as you will find in the review reports and in the annotated pdf files that the reviewers kindly provided.

You have to consider all the reviewer recommendations, taking into account that the following listed changes will be crucial to start analyzing the validity of your taxonomic and phylogenetic hypothesis.

1.-Grammar: given the language problems visualized by the reviewers, the article must be revised by a native English speaker (some suggestions of improvement are included in the annotated pdf files).
2.-Provide guarantees that the specimens that you studied are housed in a public institution and are available to other researchers.
3.-Improve substantially the item “Materials and methods” by explicating in detail the studied materials, leaving clear what of these specimens were studied by you at hand.
4.-Fgures must be improved, photographs and better quality interpretive drawings are needed and all the diagnostic characters must be easily identifiable by addition of labeling and detailed figure captions.
5.-Revise all the characters that you considered as relevant to erect a new species and include evidence enough to discard whether the differences are not taphonomic or are related to intraspecific or even ontogenetic variation.
6.-Revise the phylogenetic terminology as suggested by the reviewers and provide an independent diagnosis resultant from the revision of the putative diagnostic characters.
7.-Revise the reference list and make it consistent and adaptable to the PeerJ editorial style.
Once you have assessed all the recommended changes and concerns from the reviewers, you can submit your manuscript for a new round of revision.

Best regards,
Graciela Piñeiro
PS. I would like to specially thank very much the three reviewers for the detailed reviews on this manuscript and the useful criticism.

·

Basic reporting

Clear, unambiguous, professional English language used throughout - Professional article structure, figures, tables.

The English language is not always clear and unambiguous in this manuscript. I tried to correct it when I could (see attached doc file), but it is still in need of a linguistic review by an English speaking paleontologist. The language is professional, but sometimes the terminology is not the appropriate one (see for example Edmund, 1969 for the terminology relative to the dentition and related structures). EDMUND A. G. (1969) - Dentition. In GANS, C. & PARSONS, T. S. (eds.), Biology of the Reptilia, 1:117-200, Academic Press, London and New York.
Figures show some flaws and their number should be increased (see below).

Intro & background to show context. - Literature references, sufficient field background/context provided

The Introduction gives sufficient field background. However, it is rather chaotic, sometimes redundant and with parts that can be eliminated. I restructured it in the revised text that is attached as doc file.
A small part of the Introduction was moved into the Material and Methods section, because the latter lacked any reference to the Materials.

Literature well referenced & relevant.

The authors cite a lot of papers. Some citations are unnecessary and should be eliminated (e.g., that of Seeley, 1901 at line 343). It is surprising that more recent papers about the age of the Jehol Group formations are not mentioned in the Introduction (e.g, Chang et al., 2009).

Structure conforms to PeerJ standards, discipline norm, or improved for clarity.

Apparently, the authors followed the general PeerJ standards about manuscript structure, but did not care about details of PeerJ standards (e.g., how to write a reference list, how to write figure captions, where reporting grants etc.). This is annoying for a reviewer.
In the References, papers must be listed according to the author names in alphabetic order.
The reference list is written without following any standard or rule and contain a lot of mistakes.

Is the Beipiao Pterosaur Museum a public repository? Most journals do not accept papers about fossils that are deposited in private collections or similar institutions, mainly if they are holotypes.

Figures are relevant, high quality, well labelled & described. - Professional article structure, figures, tables.

The figures must be improved, mainly those showing fossils. A general problem is the poor quality of the drawings that does not allow recognizing many important details reported in the text. Below, there is a brief list of suggested improvements.

Figure 1. Many skeletal elements (e.g., the premaxilla, squamosal, quadratojugal, odontoid, the lower temporal fenestra, the cervical vertebrae 6 and 7, etc.) that are mentioned in the text are not indicated in figure 1. All identified bones and structures that are mentioned in the text must be indicated in this fundamental figure. Teeth and tooth positions must be indicated and numbered.
The boundaries of quadrate, quadratojugal, jugal and squamosal, that between lacrimal and jugal and those among the cervical vertebrae (mainly cervicals 6 and 7) must be drawn.
Portions of the skeletal elements that are represented only by the print of the bone (e.g., the ceratobranchial I) must be marked by dashed lines.
The beginning and the end of the prid must be clearly identifiable; mark it with arrows. The same should be done for the "dentary sulcus" in the mandibular symphysis.
According to PeerJ rules, the two parts of the figure must be indicated as A and B.

Figure 2. "Fossil position" can be eliminated in the figure. It is sufficient to write in the caption that the star indicates the site where the fossil was found.

Figure 3. The resolution of this figure is very low in the pdf I received. It is 137 kb, well below the 900 dpi that is the minimum size for a PeerJ figure. I do not understand how this could be possible. When I submitted to PeerJ, the system did not allow me to attach figures below 900 dpi: I had to enlarge the figure (without any real reason to do it).
Anyway, it would be wise to mark the nodes mentioned in the text and report the names of the relative clades (in the drawing or in the caption).
I also suggest the authors to run a Bremer test to see how 'resistant' the nodes are, reporting the corresponding values in this figure.
The number of this figure must be changed, because it is first mentioned in the text after figure 4.

Figure 4. Fig. 4A is redundant respect to Fig. 1 and must be eliminated. As in Figure 1, all bones and structures that are mentioned in the text must be indicated in the drawings. Tooth positions must be all numbered. The quality of the drawings must be improved.

Figure 5. The quality of the drawings must be improved (e.g., alveolar margins and crown-'root' boundary must be drawn) and the bones and structures that are mentioned in the text and are important for the distinction of the two taxa must be indicated.

Figure 6. There are no letters (A, B and C) in the three photographs.

Further figures must be added, showing clearly all the diagnostic features of the new species in comparison to the other species of the genus Nurhachius.

The numbering of the figures does not follow the order of citation in the text (see, for example, the first citation of Fig. 3 at line 303 and that of Fig. 4 at line 248).

The captions of the figures are not written following the rules given by PeerJ. They must be carefully rewritten following the rules. The captions with my revision are in the doc file of the text.

Raw data supplied (see PeerJ policy).

Yes.

Self-contained with relevant results to hypotheses.

Yes, but see other comments.

Experimental design

Original primary research within Scope of the journal.

Yes.

Research question well defined, relevant & meaningful. It is stated how the research fills an identified knowledge gap.

The paper is the description of a new pterosaur species and contain a phylogenetic analysis that gives original results in the definition of the Istiodactylidae and closer taxa. This is clearly defined, although some linguistic flaws, redundancy and the occasional use of inappropriate terminology make difficult the reading and interpretation.
More comments are in the doc file (which I had to transform into a pdf to upload it here; thus, read "doc file" as pdf, below) with my revision of the text.

Rigorous investigation performed to a high technical & ethical standard.

Yes, but the definition of the characters that are diagnostic of the new taxon must be better supported and explained. This is also the case of the revised diagnosis of Nurhachius.
Furthermore, the authors seem to confuse "diagnosis" and the "description of a node in a phylogenetic analysis", which are two things that are conceptually different.
More points and suggestions are in the doc file with my revision of the text.

Methods described with sufficient detail & information to replicate.

In general, yes. However, it is unclear which of the mentioned specimens have been studied personally by the authors and which data have been essentially taken from the literature.

Validity of the findings

Impact and novelty not assessed.

Novelty and importance of the described specimen are assessed.

Data is robust, statistically sound, & controlled. All underlying data have been provided

Data are sometimes ambiguous. Definition of the characters that are diagnostic of the new taxon as well as the revised diagnosis must be better supported and explained.
More comments are in the doc file with my revision of the text and general comments below.

Conclusions are well stated, linked to original research question & limited to supporting results.

See points above and comments in the doc file with my revision of the text.

Additional comments

4. GENERAL COMMENTS
The authors use the term "stem-group istiodactylids" throughout the text. This term is incorrect for at least two reasons:
FIRST: The terms "stem" and "crown" should be used only for taxa that have living representants. I know that they are often incorrectly used otherwise, but see: Jefferies, R.P.S. (1979). The origin of chordates – a methodological essay. In M.R. House (ed.). The origin of major invertebrate groups. London ; New York: Academic Press for The Systematics Association. pp. 443–447.
Pterosaurs are totally extinct, thus "stem" and "crown" cannot be used for them.
SECOND: Even if the use of the term "stem" were correct, Haopterus gracilis, Hongshanopterus lacustris and Archaeoistiodactylus linglongtaensis would not be "stem-group istiodactylids" but the stem-taxa of the clade Archaeoistiodactylus linglongtaensis + Haopterus gracilis + Hongshanopterus lacustris + Istiodactlylidae. Anyway, that would be a purported clade, because Archaeoistiodactylus linglongtaensis was shown to be even outside the Pterodactyloidea.

It is better to mention Haopterus gracilis, Hongshanopterus lacustris and Archaeoistiodactylus linglongtaensis as "taxa reported as close to the Istiodactylidae" here. Haopterus gracilis and Hongshanopterus lacustris are "taxa closer to the Istiodactylidae according to the phylogenetic hypothesis produced by the authors and Hongshanopterus lacustris is the sister-taxon of the Istiodactylidae.

As it is reported below in the text, Istiodactylidae are defined by Andres et al. (2014) as "the least inclusive clade containing Istiodactylus latidens Seeley and Nurhachius ignaciobritoi. It should be specified at the beginning of the paper.

Lines 96-99 - No, this is not necessarily the case. It depends upon the local sedimentation rates and the presence or not of hiatuses within the sections. In fact, both formations have dramatically different thicknesses in different localities. Since the thicknesses of the formations are locally very variable, comparisons should be done only within a same local stratigraphic section, anyway.

Line 104 - This section is called Material and Methods because you must list here the materials that are the subject of your study. You did not list anything.

Line 105 - You must report here which specimens you studied personally and which other were coded or described only based on the literature.

Line 143: Why the specific name is N. luei and not N. lüi? You should explain it.

Line 146: Where is this Museum? If it is in the town of Beipiao, you must write: Beipiao Pterosaur Museum, Beipiao, Liaoning Province, China. Is this museum a public istitution?

Line 151: This should essentially be a differential diagnosis: a list of features distinguishing N. luei from L. ignaciobritoi. Is it? Why do you define it "a combination of features"?

Lines 168-169: I cannot see this feature in figure 1. Please draw better it and indicate with arrows where it begins and where it ends.

I suggest to read Edmund (1969) as for the terminology to use when dealing with dentition and related structures. EDMUND A. G. (1969) - Dentition. In GANS, C. & PARSONS, T. S. (eds.), Biology of the Reptilia, 1: 117-200, Academic Press, London and New York.

Nurhachius igniaciobritoi - Longchengpterus zhaoi
The holotype of Longchengpterus zhaoi should coded separately from the holotype of Nurhachius igniaciobritoi in the phylogenetic analysis to test whether the two specimens fall as sister taxa or not. This would be pivotal is assessing whether the two specimens belong to a same species.
Supporting as much as possible with unambiguous evidence the belonging of the holotype of Longchengpterus zhaoi to Nurhachius igniaciobritoi is fundamental in the subsequent comparison with the specimen here described (BPMC-0204) and its attribution to a distinct species.

DIAGNOSTIC FEATURES OF NURHACHIUS (lines 132-138)
Lines 132-138: A diagnosis and the description of a node in a phylogenetic analysis are two things that are conceptually different. A phylogenetic analysis gives a phylogenetic hypothesis based on the data (always partial) reported in the data matrix and does not include all the known taxa. A phylogenetic hypothesis is destined to change by definition. The diagnosis is a taxonomic tool to unambiguously distinguish the taxon from all other similar taxa and must be as stable and definitive as allowed by the fossil record.
Rewrite the diagnosis listing the combination of characters that is diagnostic of Nurhachius. This means a combination of features that are all shared by N. igniaciobritoi and N. luei and that does not occur in any other pterosaur as such a combination. It should be reported when the feature is unknown in one or more of the taxa belonging to the Istiodactylidae (i.e., it is ambiguous).

Comments on characters
<<Slight dorsal deflection of the palate*>>

It is not the whole palate, it is the palatal surface of the tip of the snout, if I understood correctly.

<<Lower temporal fenestra slit-like>>

You call otherwise this fenestra in the text. Can you reliably identify the shape of this opening in N. igniaciobritoi? Based on your Fig. 4, it seems not the case.

<<dentary mandibular symphysis with gradual taper of the lateral margins>>

In which view? If you mean "in dorsoventral view", I think that this is the plesiomorphic condition within the Pterosauria.

<<triangular, laterally labiolingually compressed teeth tooth crowns lacking carinae>>

Labiolingually compressed tooth crowns is a synapomorphy of Hongshanopterus lacustris+Istiodactylidae. Thus, the diagnostic feature is only " tooth crowns lacking carinae".

<< anterior mesialmost teeth relatively longer than others>>

"Relatively longer" is a rather ambiguous definition. Anyway, this feature is plesiomorphic within the Pterosauria.

<< crowns with both labial and mesial slight concavities*>>

I do not understand this feature. In the description of the teeth of N. luei, the authors write "The lingual surface of the crown is concave with a well-marked longitudinal depression", which is not the same. The authors must clarify this point and consider the possibility that depressions could be caused by the collapse of the pulp cavity.


DIAGNOSTIC FEATURES OF THE NEW SPECIES (lines 151-156 and 379-388)

<<The quadrate is inclined at 150° instead of the 160° of N. igniaciobritoi .>>

It is a difference of only 10°. Is it really meaningful? Can you exclude it is caused by crushing or slight disarticulation? How is the slope variability within a sample of pterodactyloids belonging to a same species (e.g., Pterodactylus antiquus and Pteranodon longiceps)? Is it higher than 10°, thus the variability you observed in the three Nurhachius specimens may be intraspecific?

<< The medial process of the pterygoid, is broad and plate-like, whereas it is reduced in N. igniaciobritoi >>

Show it in a figure. Can you exclude that it is reduced in N. igniaciobritoi because it is broken?

<<The dentary median dorsal sulcus of the mandibular symphysis extends until the first pair of dentary teeth, whereas it reaches the sixth pair of dentary teeth in N. igniaciobritoi>>

Show it in a figure. Can you exclude that the difference is of taphonomic origin?

<< The odontoid (pseudotooth) lacks a lateral foramen, whereas the foramen is present in the referred specimen of N. igniaciobritoi (see Martill, 2014, fig. 7C-D).>>

That foramen is on the left lateral side in N. igniaciobritoi. Can you see the left lateral side of the odontoid in BPMC-0204? You do not figure this important feature, thus I cannot know it. Show it in a figure.

<<The odontoid has a smooth blunt occlusal surface, whereas the surface is sharp in N. igniaciobritoi >>

Can you see the occlusal surface of the odontoid in BPMC-0204? The occlusal surface should be the apex, if the odontoid is dorsally directed. You do not figure this important feature (which is not indicated in Figure 1), thus I cannot know it. I suppose it should be blunt (which is the opposite of sharp), but it is necessary to add a photograph of the structure to show it.

<<The odontoid is dorsally directed, whereas it is slightly anterodorsally directed in N. igniaciobritoi (but see Martill, 2014, p. 57, right column, lines 21-23)>>

Again, you must show it in a figure. You must prove that there is a substantial difference between the direction of the structure in N. igniaciobritoi and in BPMC-0204.

<The ceratobranchial I of the hyoid apparatus accounts for 60% of the mandibular length, whereas it accounts for 60% of the mandibular length in N. igniaciobritoi.

Actually, this does not appear to be a diagnostic feature in the Discussion section, because both species result to have the same relative length (cf. lines 382 and 387). Is it a typo?

<<12 tooth positions in each side of the upper jaw; 11 tooth positions in each side of the lower jaw.>>

This is not reported as a diagnostic feature in the Discussion section. Comparison with tooth count of N. igniaciobritoi is not discussed.
Consider that tooth count can depend upon the growth stage of the individual and is often intraspecifically variable (see Edmund, 1969). Does the tooth count variability within the sample of a pterodactyloid pterosaur represented by an adequate number of specimens support your statement? A difference of 2-3 tooth positions could be not significant.

<< The mandibular teeth extend distally beyond the symphysis, whereas they are confined to the symphysis in N. igniaciobritoi>>

Does the variability in the distalmost teeth positions within the sample of a pterodactyloid pterosaur represented by an adequate number of specimens support your statement?

Lines 389-90: <<both specimens of Nurhachius ignaciobritoi come from the upper part of the Jiufotang Formation>>.

This must be demonstrated.

REFERENCES
The reference lists is not written following the rules given by PeerJ. It is a chaotic list of citation written following different styles and full of mistakes. I tried to correct part of the mistakes but I did not do all the job. The authors must rewrite it in a correct way.

Lines 64-77 - It surprising that the references regarding the dating of the Jehol Group units are quite old and more recent papers (like Chang et al., 2009) are not mentioned.

Line 64 - Gradstein et al. (2004) is an old and partly obsolete reference. The International Chronostratigraphic Chart of ICS (IUGS) is updated twice per year. See www.stratigraphy.org
* * *
CUSTOM CHECKS

New Peerj species policies
- The name of the new taxon was not mentioned in the title.
- The new species was probably entered in Zoobank because it has a LISD, but I cannot check whether it is correct or not, of course (searching for it by Google does not give any result).

New species checks
Do you agree that it is a new species?
I agree that it may be a new species, but the authors must support their statements in a more clear and exhaustive way. This is explained in the General Comments and in the doc file with my revision of the text. Diagnostic features must also be explained by detailed figures. The differences with the other taxon/taxa must be clearly shown to the reader.

The authors affirm that "both specimens of Nurhachius ignaciobritoi come from the upper part of the Jiufotang Formation". This is an important information, because the two purported species would result to have lived in two distinct intervals of the geological time that are separated by some million years (as underlined in the Introduction). However, Authors do not give any evidence or reference to support their statement. The stratigraphic provenance of the two specimens of N. ignaciobritoi must be demonstrated.

Is it correctly described e.g. meets ICZN standard?
I discussed this in the doc file with my revision of the text. The new species is correctly described, but the emended diagnosis of Nurhachius must be reconsidered. A diagnosis of a taxon and the description of a node in a phylogenetic analysis are two things that are conceptually different.

Raw data check - Review the raw data.
If you consider the data matrix of the cladistic analysis as "raw data", it is impossible to review all the characters, character states and character state codings in only 10 day. I did not do it. Anyway, the data matrix was not produced ex novo for this paper, but was taken from a previous one. Therefore, fact that the terminology used in the character and character state definitions is sometimes inappropriate, is not a fault of the authors of the manuscript under review. As for the codings of the newly added taxa, their correctness is a personal responsibility of the authors. I never saw those specimens personally, therefore I cannot judge the correctness of the codings.

Running the data matrix gave the same results as that obtained by the authors.

Image check - Check that figures and images have not been inappropriately manipulated.
The figures do not seem to have been manipulated, but they contain some mistakes and can be substantially improved (also graphically).

·

Basic reporting

Despite the fact that I' not a native English speaker myself, I was able to detect several grammatical errors and stylistic issues. The text seems to be written by more than one author with completely discrepant styles, and some excerpts are rather confusing (see the annotated PFD file I provided). Stylistic standardization will improve clarity and is extremely recommended in this case. I also recommend a careful review by a native English speaker prior to publication.

The literature references are adequate and the authors provide an up-to-date background for the accessed issues.

Raw data was provided by the authors. Figures are competent (with the exception of Figure 3, which came to me in a exceptionally poor quality). They are, however, insufficient to support the conclusions drawn out by the authors, as some key features of the new specimen lack proper illustration.

The study is self-contained, all the results being relevant to the hypotheses.

Experimental design

This is an original research, based on a specimen never before adequately studied and illustrated. The primary research question (whether or not the new specimen represents a new taxon) is relevant and well defined. The author hypothesis was properly tested by anatomical comparisons and a robust phylogenetic analysis. Methods are properly described and the provided raw data enables replication.

Validity of the findings

The author hypothesis was duly tested with the use of adequate methodology, making the results sound, so that I was convinced of their main statements. Some relevant information lacks proper illustration (see the annotated PDF file).

Additional comments

This fine MS describes a new Chinese istiodactylid based on a wonderfully-preserved specimen, adding information on this curious pterodactyloid clade. The MS has several merits justifying its publication in PeerJ, not the least of them being the resolution of the phylogenetic position of the enigmatic taxon Archaeoistiodactylus linglongtaensis.

In spite of that, I have several concerns that must be accessed by the authors before a final version is ready for publication. All of them are detailed in the annotated .pdf file I provide together with this report.

Reviewer 3 ·

Basic reporting

no comment

Experimental design

no comment

Validity of the findings

no comment

Additional comments

This manuscript describes a unique specimen that is of great value to pterosaur specialists. It has been enjoyable to read it and review the phylogenetic analysis, with which I agree. The revision of the Istiodactylidae lineage was necessary as the authors detailed, but even those taxa close related with the istiodactylids following the phylogenetic definition established by Wang et al. (2005).

My primary concern is on phylogenetic concept of “stem group”, which should be avoided. The concept “stem group” is strictly related with “crown group”, which was introduced for classifying living organisms and their extinct relatives (Jefferies, 1979; Willmann, 2003). Thus, a stem group is used for those extinct relatives more closely related to the crown group than to any other extant organisms, but excluding the crown group (i.e., the most recent common ancestor of the living lineage and its descendants). Due to Istiodactylidae is an extinct lineage, the proposed term “stem-group istiodactylids” makes no sense. It could be substituted by a statement as “lanceodontians close to istiodactylids than anhanguerians”, but I really encourage the authors to propose a taxonomic definition of this lineage, either from Haopterus or even all the lineage from Lonchodraco+Ikrandraco.

Also, please take into account italics and punctuation marks (see below).

I have noted a few additional issues:
Line 36 – use italics in the Latin phrase “et al.” and delete comma after “Lü”
Line 37 – I recommend to write “Istiodactylus sinensis” instead of “I. sinensis”
Lines 38-39 – avoid to use the term “stem-group istiodactylids”
Line 57-59 – use italics in the Latin phrase “et al.”
Line 65 – the word “old” should be deleted
Line 87 – “and also the bird Jeholornis” should be added a reference
Line 109 – if you used the last version of TNT, you should referred as TNT 1.5 (Goloboff & Catalano, 2016)
Line 167 – “2019” instead of “2016”
Lines 440-441 – “Haopterus was recovered as a basal ornithocheiraean” following Andres et al. (2014) definition (i.e., the least inclusive clade containing Ornithocheirus simus and Anhanguera blittersdorffi) this is not true. In Holgado et al. (2019), Haopterus was recovered close but out of Ornithocheirae. A proposal of how it should be rewritten: Haopterus was recovered close to anhanguerians than istiodactylids.

Additional references
Goloboff, P. A., Catalano, S. A. (2016). TNT version 1.5, including a full implementation of phylogenetic morphometrics. Cladistics 32(3), 221–238.
Jefferies, R.P.S. (1979). The Origin of Chordates: a methodological essay. Academic Press for the Systematics Association, London. pp. 443–447.
Willmann, R. (2003). From Haeckel to Hennig: the early development of phylogenetics in German-speaking Europe. Cladistics 19, 449–479.

---

## Round 0.2 · Minor Revisions

Dear authors,

I sent your revised version of the manuscript to a new reviewer because I realized that some of the issues requested to be changed or concerns from at least one of the previous reviewers were not completely assessed. We have now an independent review that I would like that you follow to make the last modifications and improvements that your manuscript needs to be acceptable for publication in PeerJ. So, please, pay careful attention to the annotated pdf file that the reviewer kindly provided.

With my best regards,
Graciela

·

Basic reporting

This is a rereview of a major revision. The authors have corrected the points, etc. pointed out by previous reviewers. I have noted minor errors and other points that could be corrected, but ms. is publishable with minor revision.

See my detailed comments and marked up ms. in the attached pdf.

Experimental design

OK

Validity of the findings

OK

Additional comments

See my detailed comments and marked up ms. in the attached pdf.

---

## Round 0.3 · Minor Revisions

Dear authors,

I am glad to see that you have considered the commentaries and suggestions from the last reviewer and modified the manuscript accordingly. Therefore, I think that this manuscript is almost ready to be accepted for publication in PeerJ.

Please, fix the problems that I have perceived during the careful revision of the last version (included in the attached annotated pdf), which are mainly related to the reference list and citations within the text, and resubmit your manuscript.

With my best regards,
Graciela Piñeiro

---

## Round 0.4 · accepted · Accept

Dear authors,
From my own consideration, you have completed the steps requested for publication of your manuscript on a new istiodactylid pterosaur from the Early Cretaceous of China, in PeerJ. I am glad and grateful for to have had the opportunity of handle this interesting article.
Congratulations!
Best wishes,
Graciela Piñeiro

#